# Pt-Coated Ni Layer Supported on Ni Foam for Enhanced Electro-Oxidation of Formic Acid

**DOI:** 10.3390/ma16196427

**Published:** 2023-09-27

**Authors:** Antanas Nacys, Dijana Simkunaitė, Aldona Balciunaite, Ausrine Zabielaite, Daina Upskuviene, Ramunas Levinas, Vitalija Jasulaitiene, Vitalij Kovalevskij, Birute Simkunaite-Stanyniene, Loreta Tamasauskaite-Tamasiunaite, Eugenijus Norkus

**Affiliations:** Center for Physical Sciences and Technology (FTMC), LT-10257 Vilnius, Lithuania; dijana.simkunaite@ftmc.lt (D.S.); aldona.balciunaite@ftmc.lt (A.B.); ausrine.zabielaite@ftmc.lt (A.Z.); daina.upskuviene@ftmc.lt (D.U.); ramunas.levinas@ftmc.lt (R.L.); vitalija.jasulaitiene@ftmc.lt (V.J.); vitalij.kovalevskij@ftmc.lt (V.K.); birute.simkunaite@ftmc.lt (B.S.-S.); loreta.tamasauskaite@ftmc.lt (L.T.-T.)

**Keywords:** platinum, nickel foam, electroless deposition, formic acid, electro-oxidation

## Abstract

A Pt-coated Ni layer supported on a Ni foam catalyst (denoted PtNi/Ni_foam_) was investigated for the electro-oxidation of the formic acid (FAO) in acidic media. The prepared PtNi/Ni_foam_ catalyst was studied as a function of the formic acid (FA) concentration at bare Pt and PtNi/Ni_foam_ catalysts. The catalytic activity of the PtNi/Ni_foam_ catalysts, studied on the basis of the ratio of the direct and indirect current peaks (*j*^d^)/(*j*^nd^) for the FAO reaction, showed values approximately 10 times higher compared to those on bare Pt, particularly at low FA concentrations, reflecting the superiority of the former catalysts for the electro-oxidation of FA to CO_2_. Ni foams provide a large surface area for the FAO, while synergistic effects between Pt nanoparticles and Ni-oxy species layer on Ni foams contribute significantly to the enhanced electro-oxidation of FA via the direct pathway, making it almost equal to the indirect pathway, particularly at low FA concentrations.

## 1. Introduction

High energy density, facile storage, operation, and transportation make formic-acid-based fuel cells rather promising for next-generation power sources, especially for small devices and portable applications [1,2,3,4,5]. The viability of direct formic acid fuel cells (DFAFCs) intensely relies on efficient formic acid or formate electro-oxidation reactions (FAO or FORs). In general, Pt- or Pd-based materials are considered to be the most suitable and advanced catalysts for the efficiency of these reactions [2,6,7]. It is well known that FAO in acidic media occurs via two different reaction pathways on Pt [4,8,9]. FA can be oxidised to CO_2_ (i) directly via a reactive intermediate (HCOOH + A → CO_2_ + 2H^+^ + 2e^−^)—dehydrogenation or (ii) indirectly via an adsorbed CO_ads_ species produced by the dissociation of formic acid (HCOOH→B→CO_ads_ +H_2_O→ CO_2_ + 2H^+^ + 2e^−^), i.e., dehydration. Recently, a new reaction pathway has been proposed, including a hydrogen electro-oxidation reaction (HOR), where the H_2_ produced is supposed to exist as a new intermediate product that is rapidly electro-oxidised to H^+^, contributing to the overall process [10].

Typically, CO_ads_ species are identified as poisoning intermediates, but the nature of the active intermediate is still a matter of debate. A formyl COOH^–^ [11] or adsorbed formate HCOO^–^ [12,13,14,15,16] is supposed to be the rate-determining species; nevertheless, in some cases, the latter is concluded to be a spectator species rather than the active intermediate [17,18,19,20,21]. In some studies, formate is considered to be a common key intermediate in both direct and indirect pathways [14,15,16,22]. Meanwhile, a three-pathway mechanism has been proposed in Ref. [17], in which weakly adsorbed HCOOH_ads_ molecules are considered to be the active FAO intermediate, with their direct oxidation to CO_2_ being the predominant pathway. FAO reaction is very sensitive to pH of the solution [23,24,25,26,27,28,29,30], composition of formic acid/formate [29,31,32,33], temperature [34], and nature of the electrode or surface structure [15,35,36,37].

The successful commercialisation of DFAFCs is largely determined by the selection of the appropriate anode catalyst. Although Pt- and Pt-based materials are widely used in commercial applications and are the most promising electrochemical catalysts, they are rare and still suffer from high cost, insufficient durability, and low performance due to the rapid deactivation of in situ generated carbon monoxide intermediates. For this reason, developing an efficient, stable, and low-cost anode catalyst is paramount. Several strategies have been pursued to reduce carbonaceous poisoning effects and improve Pt-based catalyst performance to address these requirements. They either resist CO adsorption on the Pt surface and/or facilitate oxidative removal of adsorbed CO from the Pt surface. The first approach is realised by coupling Pt with other metals such as Ni [38,39,40,41], Bi [42,43,44], Sb [45], and Rh [46] through so-called ensemble and/or electronic effects. Another approach is based on the enrichment of the surface with oxygen-containing species via the so-called bifunctional mechanism by alloying Pt, e.g., with metal oxides such as NiO_x_ [47,48,49,50], CoOx [47], Cu_2_O [51], FeOx [52], and MnO_x_ [53,54], which are characterised by their ability to allow the electrochemical dissociation of water at potentials more negative than that of bare Pt [55,56].

In order to reduce the use of Pt nanoparticles while minimising the cost of electrocatalysts for commercial applications, emerging materials with large specific areas such as porous carbon, carbon nanotubes, carbon black, doped graphene, or graphene nanosheets [38,43,56,57,58,59] are used as supports. Recently, conductive substrates such as conductive polymers have been successfully used as catalyst supports for FAO due to their porous structures and high surface area [60,61]. Alternatively, careful engineering of nanocatalysts from solid dimensions to porous nanostructures, e.g., through a simple dealloying process, could achieve large specific areas [39,57,62]. The porous structure and alloy synergy were found to provide a significant gain in the preferred dehydrogenation pathway. The use of porous structures is of interest as they can not only shift [63,64,65] but even change [62,66,67] the reaction pathway from the undesirable indirect to the preferred direct electro-oxidation pathway of formic acid.

In this context, the use of three-dimensional Ni foam with its unique architecture as catalyst support has attracted particular attention due to its low density, high thermal and mechanical stability, high electrical conductivity, large specific surface area, and ease of reactant and product diffusion.

Recently, a Pt-coated Ni layer supported on Ni foam (PtNi/Ni_foam_) has been proposed for efficient formate electro-oxidation in an alkaline medium [68]. It showed enhanced electrocatalytic activity toward formate electro-oxidation via the direct pathway in an alkaline medium, in contrast to the pure Pt electrode. As a follow-up to our previous studies [68], the behaviour of the prepared PtNi/Ni_foam_ catalyst in acidic media is presented in this study.

## 2. Materials and Methods

The Ni foam with 20 pores cm^–1^, a bulk density of 0.45 g cm^–3^, and a thickness of 1.6 mm was purchased from the supplier GoodFellow GmbH (Hamburg, Germany). The thin Ni layer was deposited on a Ni foam substrate through the use of sodium hypophosphite as a reducing agent [68]. The Pt thin layer was electroplated on Ni/Ni_foam_ using the electrolyte containing PtCl_2_(NH_3_)_2_, NH_4_NO_3_, NH_4_OH, and NaNO_2_ (pH 8) at a current density of 1 A dm^–2^ for 40 min. The electrolyte temperature was kept at 95 °C.

X-ray photoelectron spectroscopy (XPS) analysis of samples was carried out using a Kratos AXIS Supra+ spectrometer (Kratos Analytical, Manchester, UK, 2019) with monochromatic Al Kα (1486.6 eV) X-ray radiation powered at 225 W [68].

The electro-oxidation of FA was investigated using a Zennium electrochemical workstation (ZAHNER-Elektrik GmbH & Co.KG, Kronach, Germany). A conventional three-electrode cell was used for electrochemical measurements. The Ni/Ni_foam_ and PtNi/Ni_foam_ catalysts with a geometric area of 2.45 cm^2^ were employed as working electrodes. An Ag/AgCl/KCl (3 M KCl) electrode was used as a reference, and a Pt sheet with a geometric area of 4 cm^2^ was used as a counter electrode. The bulk Pt electrode with a geometric area of 1 cm^2^ was used for comparison. Cyclic voltammograms (CVs) were recorded at a potential scan rate of 50 mV s^–1^ from the open-circuit potential value in the anodic voltammetric scan up to +1.4 V unless otherwise stated in a 0.5 M H_2_SO_4_ solution containing FA concentration in the range of 0.05–0.7 M at a temperature of 25 °C. All potential values given are referred to as “Ag/AgCl”.

Before each measurement of the electrochemical CV curves, the Pt and PtNi/Ni_foam_ electrodes were pre-treated in 0.5 M H_2_SO_4_ solution in a potential window of −0.2 to 1.3 V at a potential scan rate of 50 mV s^–1^. The electrochemically active surface areas (ECSAs) of the prepared catalysts were determined by calculating the charge associated with hydrogen adsorption (210 μC cm^–2^) [69]. CVs for the oxidative CO stripping from the surface of Pt and PtNi/Ni_foam_ catalysts were recorded in N_2_-saturated 0.5 M H_2_SO_4_ at 50 mV s^–1^. CO was adsorbed in 0.5 M H_2_SO_4_ at a potential of –0.2 V for 15 min.

## 3. Results and Discussion

X-ray photoelectron spectroscopy was used to analyse the electronic state of the surface composition of the prepared PtNi/Ni_foam_ catalyst as described in our previous work [68]. Figure 1 shows high-resolution spectra of Pt 4f (a), Ni 2p (b), and O 1s (c) for PtNi/Ni_foam_.

The determined Pt 4f spectra gave a doublet of a high-energy band (Pt 4f_5/2_) and a low-energy band (Pt 4f_7/2_). Deconvolution of the latter revealed two peaks centred at 70.9 and 72.4 eV, showing that Pt is present in two different oxidation states, Pt (0) and Pt (II), indicating that the Pt species grown on the Ni/Ni_foam_ are in the metallic state and PtO or Pt(OH)_2_, respectively [70]. The Ni 2p_3/2_ XPS spectrum split into three resolved peaks centred at 852.3 eV, 853.9, and 855.8 eV corresponds to the presence of Ni, NiO, and Ni(OH)_2_ species on the Ni_foam_ surface, respectively. The resulting XPS spectrum of O 1s is split into three resolved peaks centred at 529.8, 531.3, and 532.8 eV. The lowest energy contributions at 529.8 and 531.3 eV were assigned to the oxide/hydroxide species such as NiO and Ni(OH)_2_, respectively [71]. Meanwhile, the highest binding energy value at 532.8 eV is generally associated with physically adsorbed water molecules [72,73].

The electrochemical behaviour of bare Pt and PtNi/Ni_foam_ electrodes toward the electro-oxidation of formic acid in an acidic medium was evaluated using cyclic voltammetry. The CVs of the bare Pt and PtNi/Ni_foam_ electrodes in 0.5 M H_2_SO_4_ solution, measured at a potential scan rate of 50 mV s^−1^, are shown in Figure 2. The typical behaviour of bare Pt in acidic media is characterised by three clearly identifiable peak pairs labelled I/I’, II/II’, and III/III’. The first two pairs in the negative potential region correspond to the adsorption/desorption of hydrogen. The third, at more positive potentials, corresponds to the surface redox transition associated with the Pt/PtO transformation.

In the case of the Ni/Ni_foam_ electrode modified with Pt nanoparticles, an enormous increase in current is observed compared to the current values obtained for the bare Pt substrate (Figure 2). The dissolution of Ni in sulphuric acid takes place when anodic potentials are applied. Meanwhile, on the catalyst surface, (NiOH)_ads_ species are being formed. The reaction sequence in acidic media is as follows [74]:Ni+H_2_O → (NiOH)_ads_ + H^+^ + e^−^
(1)
(NiOH)_ads_ → Ni(OH^+^)_ads_ + e^−^(2)
Ni(OH^+^)_ads_ +2H^+^ → Ni(OH)_2_ + H^+^(3)
Ni(OH)_2_ + 2H^+^ → Ni^2+^ + H_2_O(4)
Ni(OH)_2_ → NiO(Ni_2_O_3_)(5)

Net reaction:Ni→ Ni^2+^ + 2e^−^(6)

It should be noted that although Ni species are very susceptible to dissolution in acidic media, the mesoporous Ni-Pt films appear to be more corrosion-resistant, especially with increasing Pt content, as discussed in Ref. [75]. Moreover, the latter simultaneously show very high activity in the redox reaction of Ni(OH)_2_⇔NiOOH in sulfuric acid [75].

The enormous increase in current on the PtNi/Ni_foam_ electrode indicates that it has a much larger surface area than the bare Pt substrate. The ECSAs of the prepared catalysts were determined from the CVs of the Pt and PtNi/Ni_foam_ catalysts recorded in a deaerated 0.5 M H_2_SO_4_ solution at a scan rate of 50 mV s^–1^ by calculating the charge associated with hydrogen adsorption (210 μC cm^–2^) [69]. For the bare Pt substrate, this value is 1.5 cm^2^, while for the PtNi/Ni_foam_ electrode, the average value is 71 cm^2^. Before each measurement of the electrochemical CV curve, the PtNi/Ni_foam_ electrode was pre-treated in the 0.5 M H_2_SO_4_ solution (as specified in the experimental part), and the ECSA was then re-evaluated.

Representative CV curves as a function of the FA concentration of 0.3, 0.5, and 0.7 M in 0.5 M H_2_SO_4_ solution for the bare Pt and of 0.05, 0.07, 0.1, 0.3, 0.5, and 0.7 M for PtNi/Ni_foam_ catalysts are plotted in Figure 3a,b, respectively. For clarity, CVs on PtNi/Ni_foam_ in the 0.5 M H_2_SO_4_ solution containing the lowest concentrations of FA, e.g., 0.05, 0.07, and 0.1 M, are shown in Figure 3c on a larger scale.

Three oxidation peaks labelled Peak (I), Peak (II), and Peak (III) are seen in the positive-going potential scan (Figure 3b,c). Moreover, there are also two oxidation peaks in the reverse negative potential scan, labelled Peak IV, followed by a relatively well-developed shoulder, labelled Peak V. The latter peak is more pronounced at lower FA concentrations of 0.05, 0.07, and 0.1 M FA for the PtNi/Ni_foam_ catalyst. The voltammograms determined do not undergo radical transformations with the formic acid concentration and are similar in shape to those typically found for the bare Pt electrode [19,36,76].

The first peak current (*j*^d^) for FAO on Pt and PtNi/Ni_foam_ catalysts is in the potential region of about 0.34 V and approximately 0.43–0.55 V, respectively, depending on the FA concentration, the values of which are summarised in Table 1 and Table 2. It is attributed to the direct electro-oxidation of FA via a reactive intermediate (formate) to CO_2_ according to the following reaction sequence [41]:

Direct pathway (dehydrogenation pathway):HCOOH + Pt → H_ads_ + COOH_ads_/HCOO_ads_ + H^+^ + e^−^ + Pt→CO_2_ + 2H^+^ + e^−^ + Pt(7)

The value of the direct current peak (*j*^d^) generated under the potential region of the anodic peak (I) shows the catalytic activity of the surface for the direct electro-oxidation of FA, whereas the second oxidation peak (II) at more positive potentials mainly corresponds to the indirect electro-oxidation of FA via adsorbed CO_ad_ oxidation to CO_2_, which is realised through the following reactions [41]:

Indirect pathway (dehydration pathway):HCOOH + Pt → Pt-CO + H_2_O (non-faradaic, poisoning)(8)
Pt + H_2_O →Pt-OH + H^+^ + e^−^ (Pt hydroxylation)(9)
Pt-OH + Pt-CO → CO_2_ + 2Pt + H^+^ + e^−^ (CO oxidation)(10)

The value of the indirect current peak (*j*^ind^) generated under the potential region of the anodic peak (II) characterises the surface poisoning by the CO adsorption process that effectively blocks the Pt surface required for the formation of OH_ad_ (via Equation (9)), which in turn is consumed in oxidising CO_ads_ to complete FAO (via Equation (10)). In general, the insufficient availability of OH_ad_ leads to the accumulation of CO_ads_ and limits the conversion efficiency of FA to CO_2_. It should be noted that direct FAO is not completely excluded and could occur in this potential region of peak (II) [17,18]. Meanwhile, the last peak (III) at the most positive potentials during the anodic potential scan of FAO is related to the formation of surface oxides.

During the negative potential scan, a number of electrochemical reactions are assumed to take place simultaneously, including the reductive dehydroxylation of the Pt surface and the oxidation of the FA by both direct and possibly indirect routes. Peak (IV) on the negative-going potential scan represents the electro-oxidation of carbonaceous species on a clean and real catalytic-activity-containing Pt surface after a partial reduction of the irreversibly formed surface oxides [77]. At the same time, the oxidation process at the shoulder marked as peak (V) at approximately 0.3 V, particularly on the PtNi/Ni_foam_ catalyst (Figure 3c), is influenced by CO_ad_ and the contribution of its oxidation [77].

The CVs presented in Figure 3 as well as the corresponding values of the current peaks in different potential regions for different concentrations of FA for Pt and PtNi/Ni_foam_ catalysts listed in Table 1 and Table 2, respectively, show that the increase in FA concentration results in higher current values defined in the potential regions of peak (II) for both catalysts and is followed by a potential shift of the current peaks to a more positive potential region, indicating that the electrode process is irreversible. In the case of the PtNi/Ni_foam_ electrode, an enormous increase in current emerges compared to the current values observed on the bare Pt substrate in 0.5 M H_2_SO_4_ solutions (Figure 3b). It is approximately 0.44, 53.1, and 72.3 times higher for 0.3, 0.5, and 0.7 M FA, respectively. Such an efficient enhancement is attributed to the volumetric mesoporous structure of the PtNi/Ni_foam_ catalyst, possessing a large specific surface area containing numerous active sites for the FAO reaction to proceed and not only on the top of the surface but in the vicinity of the substrate also.

A similar increase in current is observed in the peak potential region (I) of both catalysts. However, for the Pt catalyst, it is relatively negligible, whereas for the PtNi/Ni_foam_ catalyst, it is detected only at lower FA concentrations of 0.05, 0.07, 0.1, and 0.3 M, with the peak potential being shifted toward a more positive potential region. Further increases in the FA concentration result in a decrease in peak (I) current, indicating that FAO via the indirect pathway starts to dominate. The analysis of the ratio of the two oxidation current peaks (*j*^d^)/(*j*^nd^) determined for the Pt electrode shows a decrease in value from 0.18 to 0.12 with the change in the formic acid concentration from 0.3 to 0.7 M (Table 1), denoting the gain in the poisoning level of the catalyst and indicating a rather low catalytic activity toward FAO via the direct route. A low number of free Pt active sites are available for FAO via the dehydrogenation pathway (Equation (7)). The poor electro-oxidation of FA during the positive potential scan and the susceptibility of the Pt surface to CO_ad_ poisoning is confirmed by the low value of another ratio of the direct current peak value (*j*^d^) on the positive-going potential scan to the backward-going current peak value (*j*^b^) generated under the potential region of the anodic peak (IV), denoted as (*j*^d^)/(*j*^b^), which is only about 0.05.

In contrast, upon modifying Ni/Ni_foam_ with Pt particles, higher or equal current values are defined for the first current peak (*j*^d^) when increasing the FA concentration to 0.3 M (Figure 3b and Table 2), meaning that less CO is formed on the modified surface. The ratios of the current values (*j*^d^)/(*j*^ind^) in the potential regions of peaks (I) and (II) are 0.98, 1.14, and 1.00 for 0.05, 0.07, and 0.1 M FA, respectively, indicating that the contribution of the FAO by the direct route is equal to or sometimes exceeds the contribution of FAO by the indirect route on the PtNi/Ni_foam_ catalyst. Considering the margins of error, it is safe to say that these two processes are at least equal at the above concentrations. This is in sharp contrast to the processes observed at higher FA concentrations on PtNi/Ni_foam_ as well as on the bare Pt catalyst, where the indirect FAO route dominates. From this point of view, the use of the PtNi/Ni_foam_ catalyst clearly shows that FAO is intensified by the direct route at low FA concentrations. Moreover, the ratio determined is approximately 10 times higher for the PtNi/Ni_foam_ catalyst as compared to that for the Pt surface. However, with increasing FA concentration from 0.3 to 0.5 and 0.7 M, this ratio decreases from 0.75 to 0.13 or even 0.07. This shows that the level of the PtNi/Ni_foam_ catalyst poisoning increases due to the accumulation of incompletely oxidised carbonaceous species, indicating a change in the dominant pathway of the FAO reaction.

Similarly, the ratio value of (*j*^d^)/(*j*^b^) in the potential region of peaks (I) and (IV) also decreases from 0.65 to even 0.07 for FA concentrations growing up from 0.05 to 0.7 M, implying the cumulative poisoning of the PtNi/Ni_foam_ catalyst. The measurements show a higher electrocatalytic activity of the PtNi/Ni_foam_ electrode toward FAO and a significantly better tolerance of the catalyst to poisoning species, especially at lower FA concentrations, as compared to the catalytic response of the bare Pt electrode, indicating the synergy between the embedded Pt and Ni layer on the porous structure of the Ni_foam_ substrate [38]. The presence of Ni species could avoid the accumulation of carbonaceous species, especially at low FA concentrations, providing more electrochemical active sites of Pt for FAO through the direct pathway.

In order to evaluate the electrocatalytic activity of the investigated catalysts toward FAO, the current density values were normalised concerning ECSA for each catalyst in acid media (Figure 4). For the sake of simplicity, only positive-going scans are presented. These values represent the specific activity of the catalysts.

The CVs clearly show that the current density values at both potential peaks (I) and (II) for the PtNi/Ni_foam_ catalyst are significantly increased as compared to the current density values of the bare Pt electrode in the same potential region for all FA concentrations studied. In the case of 0.3 M FA, this value for the PtNi/Ni_foam_ catalyst is 5.5 times higher than the (*j*^d^) for the bare Pt catalyst and is followed by an onset potential shifted to a more negative potential region. Such efficiently improved results are attributed to the synergistic effect between Pt and the Ni layer-coated porous structure of Ni_foam_ substrate that could avoid the accumulation of incompletely oxidised carbonaceous species (CO_ads_), directing the FAO reaction toward the dehydrogenation pathway.

A comparison of the electrochemical performance in terms of (*j*^d^)/(*j*^ind^) of the catalysts included in this study with those of Pt- and Pt-based electrocatalysts used for FAO in acidic media reported in the literature is presented in Table 3. A selection of relevant references, summarised in Table 3, clearly shows that the operating conditions, in particular the acidity of the FAO achieved by applying a suitable amount of sodium hydroxide, lead to a higher value of the (*j*^d^)/(*j*^ind^) ratio.

In most cases, a pH of 3.5 was used, at which a significant amount of FA is ionised to formate anion (about one-third), which reduces the polarisation resistance and increases the ionic conductivity of the electrolyte, as well as compressing the thickness of the diffusion layer [48,51,55,56]. Meanwhile, in the present study, the PtNi/Ni_foam_ catalyst in a highly acidic solution at pH 0.3 showed that, under certain conditions, this ratio can be achieved at around 1, indicating the predominance of the FAO direct pathway.

In order to confirm better tolerance to catalyst poisoning by adsorbed carbonaceous species on the PtNi/Ni_foam_ catalyst, CO stripping measurements were adjusted. The current values measured for each sample were normalised to the ECSAs determined from the hydrogen adsorption region. Figure 5a reveals an obvious CO_ads_ oxidation current peak at approximately 0.60 V during the positive potential scan on the bare Pt electrode in 0.5 M H_2_SO_4_. Meanwhile, this peak on the PtNi/Ni_foam_ catalyst in acid solution is shifted to the negative direction by approximately 0.32 V as compared to that on Pt and is located at 0.28 V (Figure 5b), suggesting that the PtNi/Ni_foam_ catalyst has better CO tolerance than the single-metal Pt catalyst. The promotion in the electro-oxidation of carbonaceous species such as CO to CO_2_ could be attributed to the availability of transition metal oxides such as NiO_x_, which allow the electrochemical dissociation of water at potentials more negative than that of a bare Pt [55,56].

The enhanced oxidation of FA could be explained by the presence of Ni-oxy species, which are supposed to act as catalytic mediators via the above-mentioned reaction by facilitating charge transfer during the direct electrooxidation of FA to CO_2_ while simultaneously oxidising CO at a rather low potential through the following reactions [47,55,56]:Pt ... HCOOH_ad_ + NiOOH → Pt_free_ + CO_2_+ Ni(OH)_2_ + H^+^ + e^−^(11)
Pt ... HCOO^–^_ad_ + NiOOH → Pt_free_ + CO_2_ + Ni(OH)_2_ + e^−^
(12)
and/or
Pt ... CO_ad_ + NiOOH + H_2_O → Pt_free_ + Ni(OH)_2_ + CO_2_ + H^+^ + e^−^(13)

The above-mentioned reactions show that the presence of the Ni(OH)_2_ species could be relatively successful in renewing the free and active Pt sites for further FAO by directing it via the dehydrogenation pathway, especially at lower FA concentrations. However, the dissolution of Ni species in a highly acidic solution should be taken into account. In explaining the enhanced oxidation of formic acid on a binary PtNi/Ni_foam_ catalyst, the synergy of the three necessary components, each performing a very specific function, should be outlined: Pt nanoparticles serve as the active site for FAO; Ni-oxy species facilitate the oxidative removal of carbon poisons from adjacent Pt sites, thus avoiding the accumulation of CO_ads_; and finally, Ni_foam_ provides the large surface area and high electrical conductivity required for fast electrocatalysis.

## 4. Conclusions

A novel binary catalyst—a Pt-coated Ni layer supported on Ni_foam_—has been proposed for efficient FAO in acidic media. It was found that the PtNi/Ni_foam_ catalyst has a significantly higher electrochemically active surface area compared to that of Pt, equal to approximately 71 cm^2^, and shows enhanced electrocatalytic activity toward the FAO via the direct pathway as compared to pure Pt electrodes, particularly at lower FA concentrations. The prepared PtNi/Ni_foam_ catalyst showed better CO tolerance than the single metal Pt in an acidic solution. The enhanced electrocatalytic activity was due to the synergistic effects between the Pt nanoparticles and the porous structure of the Ni-oxy species layer on Ni_foam_ with large ECSAs. It is suggested that Ni-oxy species assist in the oxidative removal of accumulated carbonaceous species from the surface and act as catalytic mediators for charge transfer in the electro-oxidation process.

## Figures and Tables

**Figure 1 materials-16-06427-f001:**
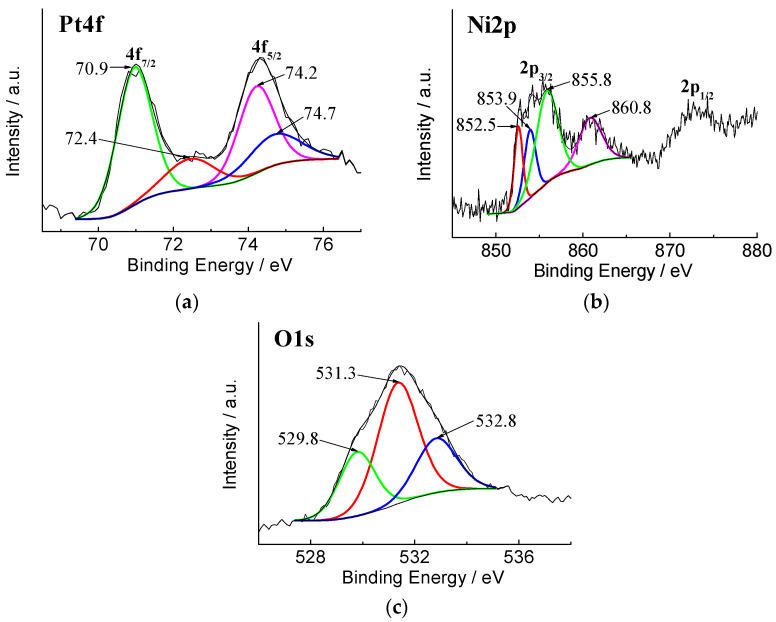
High-resolution XPS spectra of PtNi/Ni_foam_: (**a**) Pt 4f; (**b**) Ni 2p; and (**c**) O 1s [68].

**Figure 2 materials-16-06427-f002:**
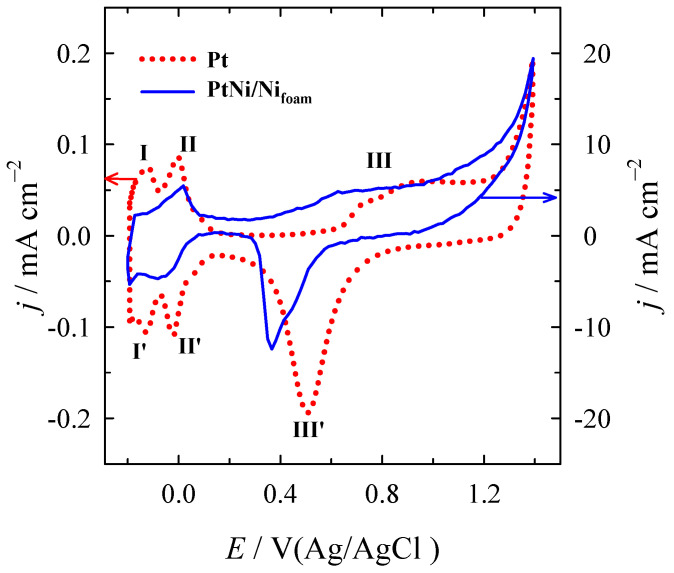
Typical stabilised CVs of Pt (dotted red line) and PtNi/Ni_foam_ (solid blue line) recorded in a 0.5 M H_2_SO_4_ solution at a scan rate of 50 mV s^−1^.

**Figure 3 materials-16-06427-f003:**
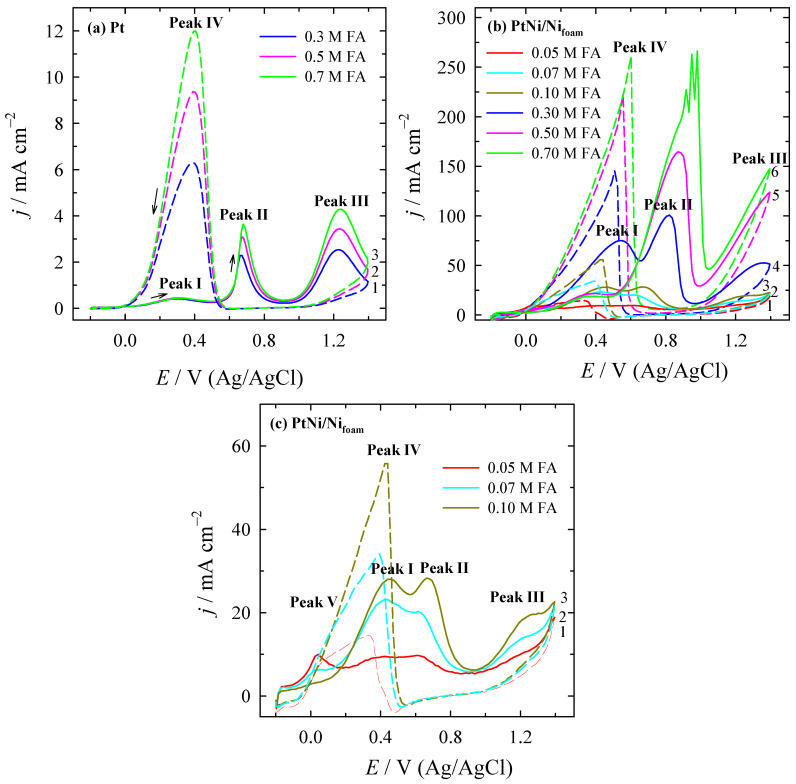
CVs of Pt (**a**) and PtNi/Ni_foam_ (**b**,**c**) recorded in a 0.5 M H_2_SO_4_ solution containing 0.3, 0.5, and 0.7 M FA; (**a**), 0.05, 0.07, 0.1, 0.3, 0.5, and 0.7 M FA; (**b**) and the same lower concentrations of 0.05, 0.07, and 0.1 M FA in the zoomed-in scale (**c**) at a scan rate of 50 mV s^−1^. (Positively going potential scan—solid lines; negatively going potential scan—dashed lines).

**Figure 4 materials-16-06427-f004:**
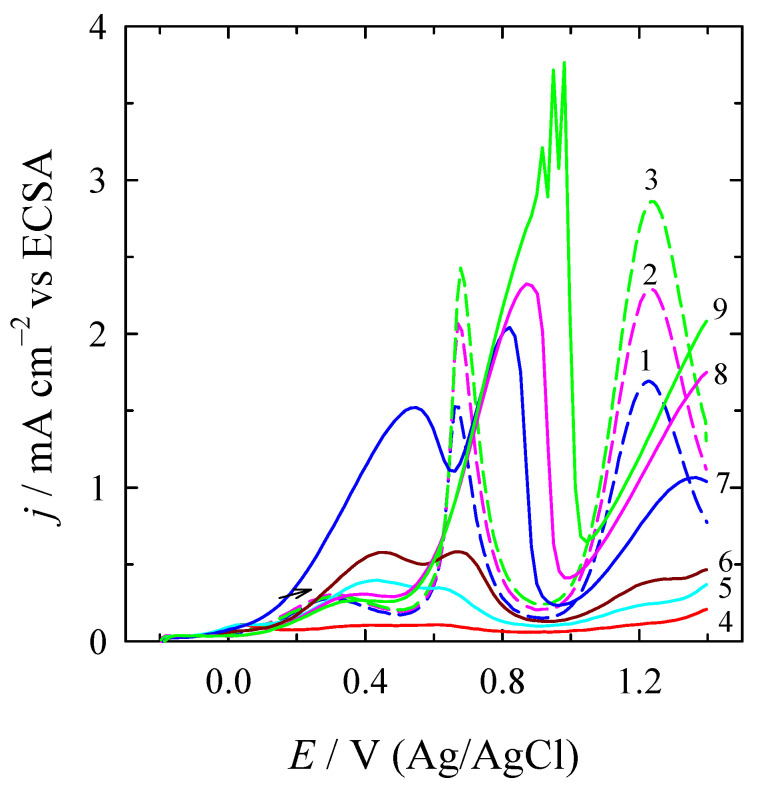
ECSA-normalised positive-going potential scans for the bare Pt catalyst, containing 0.3, 0.5, and 0.7 M FA (1–3 dotted lines), and the PtNi/Ni_foam_ catalyst, containing 0.05, 0.07, 0.1, 0.3, 0.5, and 0.7 M FA (4–9 solid lines), recorded in 0.5 M H_2_SO_4_ solution at a scan rate of 50 mV s^–1^.

**Figure 5 materials-16-06427-f005:**
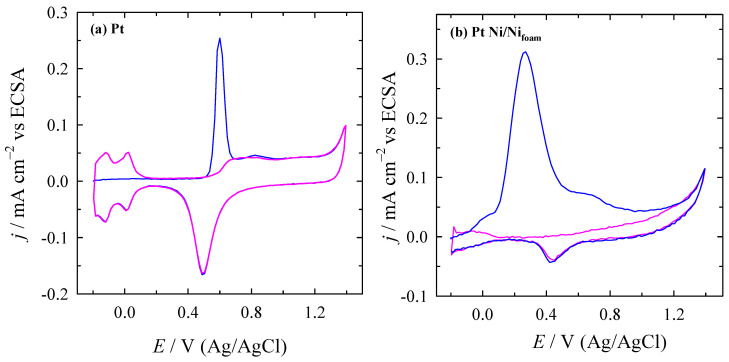
CVs for oxidative CO stripping from the surface of Pt (**a**) and PtNi/Ni_foam_ catalysts (**b**) in 0.5 M H_2_SO_4_ at 50 mV s^−1^. CO was adsorbed at −0.2 V in 0.5 M H_2_SO_4_ for 15 min; a potential sweep was carried out in N_2_-saturated solution.

**Table 1 materials-16-06427-t001:** Summary of electrochemical measurements at the Pt catalyst for the data in Figure 3a.

*c*_FA_, M	Peak I		Peak II			Peak III		Peak IV		
	*E*_pc_, V	*j*^d^, mA cm^−2^	*E*_pc_, V	*j*^ind^, mA cm^−2^	*j* * ^d^ * */j^ind^*	*E*_pc_, V	*j*^d^, mA cm^−2^	*E*_pc_, V	*j*^b^, mA cm^−2^	*(j^d^)/(j^b^)*
0.3	0.320	0.42	0.661	2.29	0.18	1.227	2.54	0.390	6.30	0.07
0.5	0.319	0.45	0.671	3.10	0.14	1.236	3.43	0.391	9.37	0.05
0.7	0.327	0.44	0.679	3.64	0.12	1.234	4.29	0.405	12.00	0.04

**Table 2 materials-16-06427-t002:** Summary of electrochemical measurements at the PtNi/Ni_foam_ catalyst for the data in Figure 3b.

*c*_FA_, M	Peak I		Peak II			Peak IV		Peak V		
	*E*_pc_, V	*j*^d^, mA cm^−2^	*E*_pc_, V	*j*^ind^, mA cm^−2^	*j* * ^d^ * */j^ind^*	*E*_pc_, V	*j*^d^, mA cm^−2^	*E*_pc_, V	*j*^b^, mA cm^−2^	*(j^d^)/(j^b^)*
0.05	0.426	9.47	0.618	9.70	0.98	0.326	14.52	0.086	8.865	0.65
0.07	0.439	23.18	0.615	20.27	1.14	0.393	34.14	0.249	25.740	0.68
0.1	0.456	28.01	0.680	28.14	1.00	0.441	55.75	0.297	38.565	0.50
0.3	0.549	75.10	0.821	100.80	0.75	0.508	145.30			0.52
0.5	0.390	21.71	0.870	164.55	0.13	0.556	223.55			0.10
0.7	0.390	18.90	0.949	263.10	0.07	0.603	261.50			0.07

**Table 3 materials-16-06427-t003:** A comparison of electrochemical performance, in terms of (*j*^d^)/(*j*^ind^), of the electrodes included in this investigation with those of Pt- and Pt-based electrocatalysts used for FAO in acidic media reported in the literature.

Catalyst	*I*^d^/*I*^ind^	Conditions of Experiment			Reference
		Electrolyte	Scan Rate, mV s^–1^	pH	
NiOx/Pt/GC	0.33	0.3 M FA	100	3.5	[58]
NiOx/Pt/CNTs/GC	∝	0.3 M FA	100	3.5	[58]
Pt/GC	0.69	0.3 M FA	100	3.5	[58]
Commercial Pt/C	0.16	0.5 M FA + 0.1M HClO_4_	50	1.0	[41]
Pt_11.1_Ni_88.9_/C	0.33	0.5 M FA + 0.1M HClO_4_	50	1.0	[41]
Pt_10.9_ Au_0.2_Ni_88.9_/C	0.34	0.5 M FA + 0.1M HClO_4_	50	1.0	[41]
Pt/C	0.29	0.5 M FA + 0.5 M H_2_SO_4_	50	0.3	[78]
Pt black	0.24	0.5 M FA + 0.5 M H_2_SO_4_	50	0.3	[78]
PtPd/C	0.87	0.5 M FA + 0.5 M H_2_SO_4_	20	0.3	[78]
Pt-TiO_x_ (700 C)	10.00	0.3 M FA	100	3.5	[79]
Pt/MWCNTs-GC	7.50	0.3 M FA	100	3.5	[80]
MnO_x_/Au/Pt/GC	30.20	0.3 M FA	100	3.5 *	[53]
NiO_x_/Au/Pt/GC	∝	0.3 M FA	100	3.5 *	[49]
nano-NiO*x*/Pt	50.00	0.3 M FA	100	3.5 *	[55]
nano-NiO*x*/Pt/GC	17.00	0.3 M FA	100	3.5 *	[56]
Au_23_/Pt_63_Co_14_	3.60	0.5 M FA + 0.1 M HClO_4_	50	1.0	[81]
PtNi/Ni_foam_	1.14	0.07 M FA + 0.5 M H_2_SO_4_	50	0.3	This work
PtNi/Ni_foam_	1.00	0.1 M FA + 0.5 M H_2_SO_4_	50	0.3	This work
PtNi/Ni_foam_	0.70	0.3 M FA + 0.5 M H_2_SO_4_	50	0.3	This work

*—pH = 3.5 + a proper amount of NaOH.

## Data Availability

Not applicable.

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
