# Peer review of "Pt-Coated Ni Layer Supported on Ni Foam for Enhanced Electro-Oxidation of Formic Acid"

_materials, 2023, doi:10.3390/ma16196427_

Round 1

Reviewer 1 Report

Overall a well planned and executed study that is reported quite well. There are a few corrections which must be done before this can be published.

1.      Materials and Methods section

All the subscripts in this section have been changed to normal text. There are also a few other examples where this has happened, for example in Eq 4 on page 4.

2.      Despite a long paragraph describing the important XPS data the spectra are not shown. They must be included in the manuscript so that readers (and especially Reviewers!!) can see the spectra and assess the evidence they present. I am particularly interested in the O(1s) spectra which purport to show adsorbed water.

3.      Figure 2c. It took me ages to work out what this chart was about. There is no explanation of it in the manuscript! The Figure legend must be expanded to explain this figure.

4.      Page 7, line 273 says: "The ratio of the current values (jd)/(jind ) in the potential region of peaks (I) and (II) equals to 0.98, 1.14, and 1.00, for 0.05, 0.07, and 0.1 M FA, respectively, pointing to the fact that FAO via the direct pathway dominates and exceeds that via the indirect route on PtNi/Nifoam"

Actually only one of those three numbers exceeds 1 and there is no indication of error. One can only conclude then that the two processes are EQUAL at these concentrations.

Author Response

Response to Reviewer 1

We are most thankful for the valuable Reviewer’s comments, which we tried to take into account. The manuscript was thoroughly revised.

Reviewer: 1.      Materials and Methods section

All the subscripts in this section have been changed to normal text. There are also a few other examples where this has happened, for example in Eq 4 on page 4.

Authors: It was corrected.

Reviewer: 2.      Despite a long paragraph describing the important XPS data the spectra are not shown. They must be included in the manuscript so that readers (and especially Reviewers!!) can see the spectra and assess the evidence they present. I am particularly interested in the O(1s) spectra which purport to show adsorbed water.

Authors: The figure with XPS data was included in the revised version of the manuscript.

Reviewer: 3.      Figure 2c. It took me ages to work out what this chart was about. There is no explanation of it in the manuscript! The Figure legend must be expanded to explain this figure.

Authors: Thank you for your comment. We have corrected the text in line with the reviewer's comment.

Reviewer: 4.      Page 7, line 273 says: "The ratio of the current values (jd)/(jind ) in the potential region of peaks (I) and (II) equals to 0.98, 1.14, and 1.00, for 0.05, 0.07, and 0.1 M FA, respectively, pointing to the fact that FAO via the direct pathway dominates and exceeds that via the indirect route on PtNi/Nifoam"

Actually only one of those three numbers exceeds 1 and there is no indication of error. One can only conclude then that the two processes are EQUAL at these concentrations.

Authors: Thank you for your pertinent comment. We have corrected the text to reflect the comment.

Reviewer 2 Report

The present work reports the Oxidation of Formic Acid in the presence of PtNi Catalyst Supported on Ni Foam

The results and discussion are properly presented. However, minor revisions should be addressed.

Question 1.

In the abstract you say « Pt-coated Ni layer supported on Ni foam catalyst…» which is correct. However, the article title starts with «PtNi Catalyst Supported on Ni Foam…». You do not support the catalyst on Ni Foam, Pt-coated Ni layer supported on Ni foam altogether are the catalyst (denoted as PtNi/Nifoam). The title should be corrected.

Also, the article title should reflect the studied process electrochemical origin, since there are catalytic processes with no current applied in formic acid oxidation. In other words, note the application, or study field.

Question 2.

It is seems a little bit strange to start the article abstract with « Currently, both formic acid (FA) and formate are of considerable interest for their possible direct production from CO2 as a green feedstock», when the article is basically devoted to the conversion of formic acid to CO2 + 2H+ via oxidation. You do not produce formic acid, it is a feedstock. Maybe the abstract should begin for example with the sentence similar to the third «High energy density, facile storage, operation and transportation make formic acid-based fuel cells rather promising for next-generation power sources, especially for small devices and portable applications. », which can highlight the importance of the process studied. What do you think?

Question 3.

Line 121 – 134. Please present in the article the XPS spectra described.

Author Response

Response to Reviewer 2

We are most thankful for the valuable Reviewer’s comments, which we tried to take into account. The manuscript was thoroughly revised.

Reviewer: Question 1. In the abstract you say « Pt-coated Ni layer supported on Ni foam catalyst…» which is correct. However, the article title starts with «PtNi Catalyst Supported on Ni Foam…». You do not support the catalyst on Ni Foam, Pt-coated Ni layer supported on Ni foam altogether are the catalyst (denoted as PtNi/Nifoam). The title should be corrected.

Also, the article title should reflect the studied process electrochemical origin, since there are catalytic processes with no current applied in formic acid oxidation. In other words, note the application, or study field.

Authors: Thank you. The title was changed.

Reviewer: Question 2. It is seems a little bit strange to start the article abstract with « Currently, both formic acid (FA) and formate are of considerable interest for their possible direct production from CO2 as a green feedstock», when the article is basically devoted to the conversion of formic acid to CO2 + 2H+ via oxidation. You do not produce formic acid, it is a feedstock. Maybe the abstract should begin for example with the sentence similar to the third «High energy density, facile storage, operation and transportation make formic acid-based fuel cells rather promising for next-generation power sources, especially for small devices and portable applications. », which can highlight the importance of the process studied. What do you think?

Authors: Thank you for your suggestion. The Introduction part was thoroughly revised.

Reviewer: Question 3. Line 121 – 134. Please present in the article the XPS spectra described.

Authors: The figure with XPS data was included in the revised version of the manuscript.